# Dark-Channel Soft-Constrained and Object-Perception-Enhanced Deep Dehazing Networks Used for Road Inspection Images

**DOI:** 10.3390/s23218932

**Published:** 2023-11-02

**Authors:** Honglin Wu, Tong Gao, Zhenming Ji, Mou Song, Lianzhen Zhang, Dezhi Kong

**Affiliations:** 1School of Transportation Science and Engineering, Harbin Institute of Technology, Harbin 150090, China; wuhonglinhit@hit.edu.cn (H.W.); 22b932003@stu.hit.edu.cn (Z.J.); 22b932004@stu.hit.edu.cn (M.S.); 2College of Electronic Science and Engineering, Jilin University, Changchun 130012, China; gaotong@jlu.edu.cn (T.G.); kongdz22@mails.jlu.edu.cn (D.K.)

**Keywords:** dark-channel soft constraints, object near-view perception, dehazing networks, feature-enhanced

## Abstract

Haze seriously affects the visual quality of road inspection images and contaminates the discrimination of key road objects, which thus hinders the execution of road inspection work. The basic assumptions of the classical dark-channel prior are not suitable for road images containing light-colored lane lines and vehicles, while typical deep dehazing networks lack physical model interpretability, and they focus on global dehazing effects, neglecting the preservation of object features. For this reason, this paper proposes a Dark-Channel Soft-Constrained and Object-Perception-Enhanced Deep Dehazing Network (DCSC-OPE-Net) for the information recovery of road inspection images. The network is divided into two modules: a dark-channel soft-constrained dehazing module and a near-view object-perception-enhanced module. Unlike the traditional dark-channel algorithms that impose strong constraints on dark pixels, a dark-channel soft-constrained loss function is constructed to ensure that the features of light-colored vehicles and lane lines are effectively maintained. To avoid resolution loss due to patch-based dark-channel processing for image dehazing, a resolution enhancement module is used to strengthen the contrast of the dehazed image. To autonomously perceive and enhance key road features to support road inspection, edge enhancement loss combined with a transmission map is embedded into the network to autonomously discover near-view objects and enhance their key features. The experiments utilize public datasets and real road inspection datasets to validate the performance of the proposed DCSC-OPE-Net compared with typical networks using dehazing evaluation metrics and road object recognition metrics. The experimental results demonstrate that the proposed DCSC-OPE-Net can obtain the best dehazing performance, with an NIQE score of 4.5 and a BRISQUE score of 18.67, and obtain the best road object recognition results (i.e., 83.67%) among the comparison methods.

## 1. Introduction

The image-based intelligent road inspection system can effectively extract road surface cracks, grooves, and other road disease objects and then provide adequate information for road management [1,2,3]. However, due to weather conditions, the haze interference phenomenon for road inspection will have a greater negative impact, which may seriously damage the image’s visual effect, thus hindering the execution of inspection work. Not only that, but haze contamination may also reduce the contrast of the images and cause confusion about objects in the road, such as crack features, and then interfere with the object extraction results obtained via intelligence methods. To recover image quality for effective road inspection, it is necessary to study the single-image dehazing method to substantially improve the quality of hazed images and ensure the implementation of intelligent road inspection under different cases and the maintenance of municipal facilities.

Currently, popular single-image dehazing methods are mainly divided into two categories: one is the classical physical or empirical-based dehazing methods, and the other is the deep learning-based dehazing methods. For the former, these methods are designed based on physical models or empirical expressions. The physical model-based methods are based on the following essential representation of the image hazing process [4,5]:(1)I(x)=J(x)t(x)+A(1−t(x))
where *I*(*x*) and *J*(*x*) denote a hazed image and haze-free image, respectively. *A* and *t*(*x*) are the global atmospheric light and transmission map. The value of the transfer map, *t*(*x*), is related to the transmission distance and the particle coefficient of the atmosphere, which can be expressed as *t*(*x*) = exp(−*β*(*x*)*d*(*x*)), where *β* is the particle coefficient of the atmosphere. *d*(*x*) as the scene depth of the image is related to the coordinates of the camera. Notice that Equation (1) is an underdetermined equation, which requires additional prior knowledge to make the equation solvable to obtain the haze-free image, *J*(*x*). To solve this problem, most physical-based methods construct constraints on the transmission map using image contrast and color differences priori to make Equation (1) solvable. The most representative classical physical dehazing method is the dark-channel prior (DCP) method [6], which is based on the statistical analysis of a large number of outdoor images. The researchers found a phenomenon that natural images generally have a certain band of luminance that is close to 0 in different image blocks. Based on this prior, the DCP method achieves image dehazing and obtains acceptable results in different datasets. To improve the thin cloud interference in remote sensing images, multiscale DCP methods were developed to utilize image dehazing by combining the low- and high-frequency components of the image [7]. In addition, using the smoothing prior to haze, researchers realized single-image dehazing using gradient response filtering of haze [8]. In addition to physical-based methods, empirical-based methods utilize the statistical rule between image features and fog depth for image dehazing. For example, by constructing the linear model to describe the relationship between the scene depth and the image color, the color attention prior (CAP)-based haze removal method is developed to achieve single-image dehazing [9]. In addition, by observing haze-free images, the researchers found that the image color can be composed of a small number of spatially dispersed clusters, while the clusters of hazed images will form “haze lines” so as to establish the non-local assumption of the image color, and then complete the image dehazing [10]. Despite the computational advantages of physical- or empirical-based methods, since these dehazing methods rely on subjective analysis or empirical assumptions, these methods fail when the analysis or assumptions are not met and produce poor results.

On the other hand, deep learning-based methods, especially for convolutional neural networks (CNNs), are widely used in image dehazing for their ability to effectively obtain discriminative features and scene semantic information. Classical deep learning-based dehazing networks are end-to-end [11,12]. For example, to achieve single-image dehazing, a feature fusion network with a multi-scale attention mechanism is constructed to extract features at different scales of the image to improve the dehazing performance [13]. In addition, to obtain a haze-free image without an atmospheric scattering model, researchers have constructed a generalized convolutional neural network model utilizing end-to-end training to achieve image dehazing [14]. A conditional generative adversarial network (cGAN) was created to perform image dehazing using an end-to-end trainable neural network through VGG features and L1 regularized loss of the gradient prior [15]. Combined with a smooth expansion technique, a multi-scale deep neural network was proposed for single-image dehazing [16]. In contrast to the end-to-end dehazing networks, another group of dehazing networks accomplishes image dehazing indirectly by estimating the transmission map. For example, task-oriented dehazing networks were constructed to utilize encode–decode modules and recurrent neural networks to simulate a multi-stage dehazing process [17]. For example, unlike traditional end-to-end networks that take haze-free images as outputs, researchers have utilized a maximal output layer to extract haze-related features and then performed end-to-end training to obtain transmission maps, which can be used to calculate haze-free images [18]. In addition, multi-scale convolutional neural networks were constructed to utilize multi-scale convolution to mine the mapping relationship between the input image and the transmission map to indirectly obtain haze-free images [19]. Further, combining dark and light channel constraints, a Principled Synthetic-to-real Dehazing (PSD) framework was established [20], which utilizes a dehazing backbone network trained with synthetic haze images to achieve rough dehazing, and then finely tunes the network with real images to improve the final dehazing results.

It is noted that the existing physical-based methods utilize ideal physical formulations that lack consideration of the complexities of road environments and objects. Take the DCP as an example. The DCP assumption can not be applied to light-colored vehicles and lane lines in road inspection images. In addition, the existing deep learning-based haze removal methods rely on a large number of synthetic haze images as training samples, and their results are susceptible to the training samples. If the synthetic haze images do not satisfy the physical priori, their results will become worse. Moreover, the existing haze removal methods are global-oriented and do not consider the objects in the image. For road inspection images, the purpose of dehazing is not only to enhance the visual characteristics but also to recover and improve the feature discrimination ability of road disease objects to ensure the effective execution of road inspection work. To integrate the interpretability of the physical-based methods and the strong learning ability of deep learning-based methods, this paper proposes a Dark-Channel Soft-Constrained and Object-Perception-Enhanced Deep Dehazing Network (DCSC-OPE-Net) for road inspection images, which is divided into two processing stages: the dark-channel soft-constrained (DCSC) image coarse dehazing module and the near-view object-perception-enhanced (OPE) module. Compared with the traditional dehazing networks that lack interpretability, the proposed DCSC module combines the dark-channel prior to make the dehazed results satisfy the physical prior of haze processing. Considering that the traditional dark-channel prior is not applicable to the light-colored vehicles and lane lines contained in the road, the DCSC loss function is constructed to process road images by maintaining the features of light-colored vehicles and lane lines. To avoid the resolution loss of key objects due to patch-based DCP processing for image dehazing, the transmission map is utilized to determine the near-view image and embedded into the proposed OPE module for strengthening the object features in the dehazed image, and then to support practical road inspection. The main contributions of this paper can be summarized as follows:(1)To realize compelling image dehazing under different complex road situations, the DCSC-OPE-Net is proposed for the haze removal of road inspection images. The proposed networks embed the dark-channel soft-constrained priori to recover the feature of different colors of objects contained in the road and learn the atmosphere transmission function to effectively recover haze-free images, which makes the results not only satisfy the physical priori but also the shortcomings of traditional DCP to adapt to white vehicles and lane lines, so as to obtain a better dehazing effect.(2)To recover and enhance the feature-discriminative ability of the road disease object, the OPE module is proposed and embedded into the proposed dehazing network. The OPE module can effectively avoid the loss of image resolution caused by patch-based DCP processing. Not only that, the OPE module can absorb the information of the scene transmission map and transform it into depth information so as to perceive and enhance the discriminative features of the near-view object of the road inspection, which ensures the effective execution of road inspection.(3)To verify the performance of the proposed DCSC-OPE-Net for road inspection images, the experiments are conducted on public datasets and road inspection images. The experimental results demonstrate that the proposed DCSC-OPE-Net outperforms typical DCP methods and state-of-the-art (SOTA) deep learning-based dehazing methods using the mainstream evaluation metrics. More importantly, the proposed DCSC-OPE-Net not only improves the visibility of the resulting image, but also enhances the discriminative ability of the objects contained in the road, which indicates better road object detection results.

The remainder of this paper is organized as follows: Section 2 introduces the classical DCP method. Section 3 gives the proposed DCSC-OPE-Net. Section 4 details the experiments that were carried out to evaluate the performance of the proposed method compared to the typical DCP method and SOTA deep dehazing network using different datasets. Section 5 gives our conclusion.

## 2. Related Work

### 2.1. Shallow Dehazing Method

To realize the dehazing of a single image, it is noted that the single image lacks the image depth and scene radiance attenuation information, and thus Equation (1) is ill posed. By observing a large number of outdoor images, the researchers found that any small patch of a natural image containing a pixel at a certain channel closes to 0. This is due to the fact that natural images are almost full of brightly colored objects, shadows, and dark object surfaces. Based on this assumption, the following equation constraint is constructed:(2)$$\underset{y\in \Omega \left(x\right)}{\min}\left(\underset{c\in \left\{r,g,b\right\}}{\min}\frac{J_{i}^{dc}\left(x;c\right)}{A_{c}}\right)=0$$
where $J_{i}^{dc}\left(x;c\right)$ and *A_c_* denote the haze-free image of the *c*th channel generated by the DCP and the atmosphere light of the *c*th channel, respectively. Based on this constraint, the transmission map can be calculated to obtain the haze-free image using Equation (1). According to Equation (2), the DCP leads the same patch, sharing the same transmission value, which thus presents the patch effect. Unlike DCP, the Laplace achieves dehazing by assuming that there exists a linear relationship between the image color and transmission map [21], while such an assumption may lead to redundant details in the dehazed results [22]. Unlike the linear assumption used to remove the patch effect, the energy minimization method is constructed to utilize piecewise smooth processing to obtain dehazing results [23]. In addition to improving the DCP in terms of the patch effect, some researchers extend the fast version of the DCP by considering the coherence between sequence images to apply in the video dehazing [24]. Furthermore, by calculating the haze abundance of different channels, the variant of the DCP method is also employed for hyperspectral images [7].

However, this constraint makes it difficult to obtain valid results for objects similar to atmospheric light, such as white vehicles and white lane lines. In addition, this patch-based dehazing method makes the same image patch share the same transmission value, which reduces the spatial resolution of the object details and affects the detailed features of the object in the images.

### 2.2. Deep Dehazing Networks

The representatives of the existing deep dehazing networks are end-to-end, which uses a large number of haze images for training to approximate the mapping from haze images to haze-free images. Different network architectures are used for this type of network design. For example, the FFA-Net utilizes feature fusion attention to fuse features from different channels and pixels to achieve image dehazing [25]. Applying adversarial learning technology, the cycle-consistent generative adversarial networks are constructed for single-image dehazing [26]. Additionally, the adversarial learning technology is also combined with attention mechanism [27] or ensemble learning [28] architecture to achieve image dehazing [29,30]. Furthermore, unlike the dehazing methods that are trained by outdoor data only, the PDR-Net uses indoor and outdoor images to train the networks effectively [31]. The drawback of this type of network is that their results lack interpretability because they ignore the haze processing equation. For this reason, researchers build networks that first estimate the transmission map and then compute the haze-free image [17,18,19,20]. Although this network possesses the interpretability of the haze process, it is difficult to ensure that the estimated transmission map is reasonable due to the ill-posed property of Equation (1). It is worth noting that the PSD approach, which embeds the network in a prior of light and dark channels, can inherit the validity of the physical model. However, the loss of spatial detail information through patch-based processing has not been resolved.

## 3. Deep Dehazing Network for Road Inspection Images with Dark-Channel Weak Constraints and Object Perception Enhancement

To effectively remove the haze of road inspection images and enhance the road object feature by integrating the interpretability of the physical model and the learning ability of deep learning methods, this paper proposes the DCSC-OPE-Net for the haze removal of road inspection images. The proposed method consists of a DCSC-based coarse dehazing module and an OPE-based module, as shown in Figure 1. Given haze image, $I_{i}$, the DCP is applied to generate a transmission map, $T_{i}^{dc}$, and atmospheric light, *A*. According to $T_{i}^{dc}$ and *A*, the MSBDN module embedding with DCSC loss is used to generate a coarse haze-free image, $J_{i}$. Then, the proposed OPE module is utilized to convert $J_{i}$ to $F\left(J_{i},\theta \right)$, and then utilizes near-field object edge loss to obtain the enhanced haze-free image, ${\tilde{Y}}_{i}$. The coarse dehazing module uses U-Net as the backbone network, and applies a local pixel attention and global color attention module to learn the atmospheric light transmission map and the dark channel of haze images. Then, the OPE module exploits multi-scale, multi-depth convolution operations to excavate the object’s refined features, and then employs the transmission map to percept the near-view object to improve the degradation of the object’s detailed information caused by the DCP process, and then output the final haze-free images.

### 3.1. Dehazing Backbone Network

#### 3.1.1. DCSC Module

Given the training set, ${\left\{I_{i},Y_{i}\right\}}_{i=1}^{N}$, where $I_{i}$ and $Y_{i}$ denote the *i*-th hazed image and the true value of the haze-free image, respectively, in view of the better fitting ability of deep learning-based image-dehazing methods, this paper adopts an MSBDN network [32] as the backbone of the dehazing network. The MSBDN network adopts the U-Net [33] architecture, which utilizes multi-scale feature encoding and boosted multi-scale feature decoding to enhance the dehazing effect and uses the dense feature fusion module to absorb the complementary information of multi-scale features.

To increase the performance of the MSBDN network, based on the image multi-scale feature layer, the local pixel and global color attention module are added to better balance the effects of local and global features on the dehazing results. The local pixel attention is obtained by two convolutional layers, sigmod and the RELU activation function, to obtain the weights of different pixels. For the *i*th input image, $I_{i}$, the pixel attention is obtained using the following equation:(3)$$M\left(I_{i}\right)=K\left(I_{i}\right)\times sig\mathrm{mod}\left(Conv\left(sig\mathrm{mod}\left(conv\left(K\left(I_{i}\right)\right)\right)\right)\right)$$
where $K\left(\cdot \right)$ and $conv\left(\cdot \right)$ denote the input data and the convolution operation, respectively. According to the above equation, it can be found that the local pixel attention layer uses the fusion of pixels with the features extracted via convolution to improve the implicit information of a particular pixel. The global color attention layer is calculated using the following equation:(4)$${\tilde{C}}_{n}=C_{n}\times sig\mathrm{mod}\left(Conv\left(sig\mathrm{mod}\left(conv\left(Global\left(C_{n}\right)\right)\right)\right)\right)$$
where *Cn* and $Global\left(\cdot \right)$ denote the *n*th channel and the global average pooling, respectively. The global color attention uses global pooling and deep feature extraction to obtain the weights of different channel features and then realize the effective fusion of channel features. The features extracted via the pixel attention and color attention modules are embedded into the MSBDN model, and the final output of this model can be expressed as
(5)$$D\left(E\left(I_{i};\theta_{en}\right),M\left(I_{i}\right),{\tilde{C}}_{n};\theta_{de}\right)$$
where $E\left(;\theta_{en}\right)$ and $D\left(;\theta_{de}\right)$ denote the MSBDN network encoder and decoder, respectively. By integrating pixel attention and color attention features, the backbone network outputs a rough haze-free image, ${\tilde{I}}_{i}$.

#### 3.1.2. DCSC Embedding

For the output, ${\tilde{I}}_{i}$, the conventional MSBDN network only outputs haze-free images and lacks physical interpretability. Considering the dehazing process, it has a classical atmospheric light transfer model, as follows:(6)$$I=J⊙T+A⊙\left(1-T\right)$$
where *I*, *J*, *T*, and *A* denote the input hazed image, the haze-free image, the transmission map with the value [0, 1], and the atmospheric light, respectively. The operator ⊙ is the Hadamard product, i.e., the elementwise product.

According to Equation (6), the haze image contains two components, i.e., the observed scene radiance, J⊙T, and the observed atmospheric light, $A⊙\left(1-T\right)$. To make the dehazing result physically interpretable, the DCP method is applied to obtain the atmospheric light, *A*. Then, the output layer utilizes a 3 × 3 convolution layer to obtain the transmission map, *T_i_,* and the rough haze-free image, $J_{i}$. To ensure the effectiveness of the dehazing results, the 2-norm dehazing loss function is applied to measure the difference between the true haze-free image and the output haze-free image, ${\tilde{I}}_{i}$:(7)$$L_{M}=\frac{1}{N}\sum_{i=1}^{N}{\left\|\tilde{J_{i}}-J_{i}\right\|}_{F}^{2}$$
where $\tilde{J_{i}}$ denotes the true value of the haze-free image of the *i*th training samples. For the classical DCP, the output results are needed to meet Equation (6). However, the drawback of the DCP method is that it cannot work well for white-colored objects in the road. To solve this problem, the dark-channel soft-constrained (DCSC) loss is proposed to improve the dehazing performance of the classical DCP. DCSC loss consists of two parts, i.e., DCSC reconstruction loss and DCSC residual loss. For DCSC reconstruction loss, combined with Equation (6), to ensure that the network output satisfies the atmospheric light transfer model, the intention of DCSC reconstruction loss is to make the output haze-free image satisfy the physical prior as much as possible:(8)$$L_{rec}=\frac{1}{N}\sum_{i=1}^{N}{\left\|I_{i}-\left(J_{i}⊙T_{i}+A⊙\left(1-T_{i}\right)\right)\right\|}_{F}^{2}$$

Subsequently, we will construct the DCSC residual loss. To further enhance the physical properties of the network, considering that the DCP dehazing method can be more effective in solving dehazing applications in different scenarios, the DCP model is embedded into the network as a sub-module. Using the DCP sub-module, the haze-free image, $J_{i}^{dc}$, and the transmission map, $T_{i}^{dc}$, can be obtained. Then, according to the DCP, the results need to satisfy the following:(9)$$\underset{y\in \Omega \left(x\right)}{\min}\left(\underset{c\in \left\{r,g,b\right\}}{\min}J_{i}^{dc}\left(x;c\right)\right)=0$$
where $J_{i}^{dc}\left(x;c\right)$ denotes the haze-free image of the *c*th channel generated by the DCP. Equation (9) indicates that the haze-free image generated via the DCP method must always have a patch of a certain channel within a pixel equal to 0. However, for the road inspection image, the white lane lines and white vehicles on the road do not satisfy the DCP assumption.

For the haze process (see Equation (6)), the minimization of the channel brightness of an image patch is taken on both sides, which can be expressed as
(10)$$\underset{y\in \Omega \left(x\right)}{\min}\left(\underset{c\in \left\{r,g,b\right\}}{\min}\frac{I_{i}\left(x;c\right)}{A^{c}}\right)=T_{i}\left(x\right)\underset{y\in \Omega \left(x\right)}{\min}\left(\underset{c\in \left\{r,g,b\right\}}{\min}\frac{J_{i}\left(x;c\right)}{A^{c}}\right)+1-T_{i}\left(x\right)$$
according to the DCP, i.e., $\underset{y\in \Omega \left(x\right)}{\min}\left(\underset{c\in \left\{r,g,b\right\}}{\min}\frac{J_{i}\left(x;c\right)}{A^{c}}\right)=0$. In addition, some of the objects, e.g., white vehicles and lane lines, do not satisfy this prior., i.e., $\underset{y\in \Omega \left(x\right)}{\min}\left(\underset{c\in \left\{r,g,b\right\}}{\min}\frac{J_{i}\left(x;c\right)}{A^{c}}\right)>0$. Combined with the above two mathematical expressions, we have
(11)$$T_{i}\left(x\right)=\left\{\begin{array}{l} 1-\underset{y\in \Omega \left(x\right)}{\min}\left(\underset{c\in \left\{r,g,b\right\}}{\min}\frac{I_{i}\left(x;c\right)}{A^{c}}\right)\\ 1-\underset{y\in \Omega \left(x\right)}{\min}\left(\underset{c\in \left\{r,g,b\right\}}{\min}\frac{I_{i}\left(x;c\right)}{A^{c}}\right)+\nu \end{array}\right.$$
where ν denotes an arbitrary nonzero constant. When $T_{i}\left(x\right)=1-\underset{y\in \Omega \left(x\right)}{\min}\left(\underset{c\in \left\{r,g,b\right\}}{\min}\frac{I_{i}\left(x;c\right)}{A^{c}}\right)$, i.e., $T_{i}\left(x\right)=T_{i}^{dc}\left(x\right)$, where $T_{i}^{dc}\left(x\right)$ denotes the value of the transmission map of the DCP, the model satisfies the dark-channel prior. When $T_{i}\left(x\right)=1-\underset{y\in \Omega \left(x\right)}{\min}\left(\underset{c\in \left\{r,g,b\right\}}{\min}\frac{I_{i}\left(x;c\right)}{A^{c}}\right)+\nu$, i.e., $T_{i}\left(x\right)=T_{i}^{dc}\left(x\right)+\nu$, the model does not satisfy the dark-channel prior. 

To inherit the excellent properties of the DCP method and meanwhile improve the disadvantage of the DCP, the DCSC residual loss is proposed to deal with these two cases, which encourages that most pixels of an image satisfy the traditional DCP and tolerates a small number of image patches that do not satisfy the DCP; that is, it tolerates the existence of the white vehicles and lane lines of the road inspection image. The proposed DCSC loss is as follows:(12)$$L_{res}={\left\|L*T_{i}-L*T_{i}^{dc}\right\|}_{1}$$
where operator ∗ means convolution operations. The *L* is the Laplace convolution kernel that is used to obtain the edge of the transmission map. The value of *L* is defined as
(13)$$L=\left[\begin{array}{l} 0,-1,\;0\\ -1,4,-1\\ 0,-1,\;0 \end{array}\right]$$

Applying $L*T_{i}-L*T_{i}^{dc}$, the edge difference between the transmission map generated from the proposed networks and the transmission map of the DCP can be obtained. Since the edges of white-colored objects in an image occupy a very small portion, using the L1 norm, the number of entries of $L*T_{i}-L*T_{i}^{dc}$ exceeding zeros should be as small as possible. In this way, the networks can handle a large number of dark-channel regions while accepting that small areas of white vehicles with lane lines do not satisfy the DCP. Due to the above reason, the loss is named as DCSC. Then, the output, $J_{i}$, of the DCSC module can be regarded as the coarse results of a dehazed image.

### 3.2. OPE Module

#### 3.2.1. OPE Backbone

According to Equation (14), although the DCSC module can strengthen the dehazing results, the transmission map generated by the DCSC module leads the small patch of an image to share the same transmission value, making the image patch challenging to present the detailed information.
(14)$$T_{i}\left(x\right)=1-\underset{y\in \Omega \left(x\right)}{\min}\left(\underset{c\in \left\{r,g,b\right\}}{\min}\frac{I_{i}\left(x;c\right)}{A^{c}}\right)$$

To maintain the key object detail information of the road inspection image, one can consider using super-resolution networks to enhance the object feature [34,35,36,37,38], while the existing super-resolution networks are whole-image-oriented rather than object-oriented. To enhance key road disease objects, unlike the existing super-resolution networks, the object-perception-enhanced (OPE) module is constructed to strengthen the detailed information of the haze-free image.

To make this module recover the original small-scale detail information of the road image as much as possible and maintain the large-scale contextual structure information of the road scene, considering that a single scale and depth of feature extraction often has a sense of the wild is too limited to a particular feature. This single-scale feature extraction tends to stagnate and decrease performance as the depth increases. For this reason, we propose an efficient multi-scale feature extraction method. The method not only performs feature extraction from a multi-scale perspective but also fully considers the resources and utilizes residual short connections to achieve the cross-layer reuse of multi-scale and multi-depth features. Specifically, the module first performs feature extraction with the help of four scales of convolution, then three scales, and finally two scales, which can make full use of the multi-scale technique and alleviate the computational resources. This step is known as the multi-scale residual block (MR block). Furthermore, the feature extraction capability of the model is improved by constructing different numbers of blocks. The proposed network uses 60 blocks, and the features are extracted at a deeper level through hierarchically connected blocks, which provide more texture information for subsequent reconstruction. Finally, the enhanced haze-free image output is obtained via image upsampling and convolution operation.

To make the output of the super-resolution module consistent with the real haze-free image, the mean squared error loss of the discriminator between the output haze-free image and the real high-resolution haze-free image is utilized as a loss as follows.
(15)$$L_{mse}\left(\theta \right)=\frac{1}{N}\sum_{i=1}^{N}{\left\|F\left(J_{i},\theta \right)-Y_{i}\right\|}_{F}^{2}$$
where ${\tilde{Y}}_{i}=F\left(J_{i},\theta \right)$ and $Y_{i}$ denote the *i*th haze-free image of the output with enhanced resolution and the real high-resolution haze-free image, respectively. The purpose of using this loss is to enhance the clarity of the output results to ensure the validity of the dehazing results.

#### 3.2.2. Near-View Object Perception and Feature Enhancement

For the road inspection application, the dehazing model is necessary to maintain and enhance the discriminative features of key objects on the road surface in addition to improving the global visual properties of the images. By observing the road inspection images, it can be found that the inspection images generally contain far-view areas, such as the sky and distant buildings, as well as near-view areas, such as the objects on the road surface. To strengthen the near-view object features of the road surface, this paper embeds near-view feature enhancement loss in the super-resolution module. Note that $T_{i}$ is related to the scattering coefficient of the atmosphere and the scene depth. The greater the depth of the scene, the smaller the corresponding $T_{i}$. To enhance the near-view object features, this paper utilizes the transmission map, $T_{i}$, to represent the object depth map [6]:(16)$$T_{i}\left(x\right)=e^{-\beta d\left(x\right)}$$
where β and $d\left(x\right)$ denote the scattering coefficient and the scene depth, respectively. Therefore, $T_{i}\left(x\right)$ is a monotonically decreasing function of the scene depth. The near-view object $T_{i}\left(x\right)$ has a larger value, while the far-field object $T_{i}\left(x\right)$ has a smaller value. According to Equation (16), the depth of an image is negatively related to $T_{i}\left(x\right)$. In this way, $T_{i}\left(x\right)$ can be regarded as an alternative expression of the scene depth to weigh near-view objects, i.e., the following term can be constructed to weigh the output dehazing results:(17)$$T_{i}⊙F\left(J_{i},\theta \right)$$
where $F\left(J_{i},\theta \right)$ is the output haze-free image. To strengthen the object features, the edges of the objects in the final haze-free image should be enhanced so that they are easier to be identified. Combining with the Laplace convolution kernel, *L*, the following object edge loss function is constructed:(18)$$L_{edge}\left(\theta \right)=\frac{1}{N}\sum_{i=1}^{N}{\left\|L\times \left(T_{i}⊙F\left(J_{i},\theta \right)\right)-L\times \left(T_{i}\left(x\right)⊙{\tilde{Y}}_{i}\right)\right\|}_{F}^{2}$$

The loss function of the above equation is to make the edge of near-view object similar to the true object edge. Also, to avoid the image edges being too sharp and lacking realism, a histogram equalization operation is applied to obtain the final haze-free image.

## 4. Experimentation and Analysis

To validate the effectiveness of the proposed DCSC-OPE-Net for haze removal in road inspection images, the publicly available dataset and the road inspection image dataset are used to validate the proposed method compared with SOTA methods under different evaluation metrics, so as to verify its ability to effectively restore the quality of road inspection images and ensure the extraction and analysis of key road objects.

Details of the datasets

The datasets in this paper are divided into two parts. One is the OTS (outdoor training set) of the publicly available RESIDE dataset. This dataset includes 2061 real outdoor images from real-time weather in Beijing, which consist of paired haze-free outdoor images and haze-generating images. 

The other one is the road inspection dataset constructed in this paper. The road inspection dataset contains the real road inspection data of Shenzhen, which has completed the inspection images acquired from road inspection vehicles of more than ten roads, including Binhai Avenue, Shennan Avenue, Yuewan Avenue, Shahe West Road, Baishi Road, and so on. The dataset consists of 6264 inspection images, which contain clear outdoor images and synthetic haze images, as well as 200 pairs of real haze and haze-free images acquired from the same road sections.

Evaluation metrics

In our experiments, the following metrics are used for the evaluation of the dehazing results. 

The BRISQUE (Blind/Referenceless Image Spatial Quality Evaluator) and NIQE (Natural Image Quality Evaluator): these two types of metrics are commonly used image quality evaluation metrics based on pixel statistics results. These two metrics, as hybrid indicators, include different evaluation manners, such as the gradient, contrast, color distribution, etc. Meanwhile, they do not need reference images and can automatically evaluate the quality of images, which is widely used in the evaluation of dehazing methods [39].

Mean Gradient (MG): This metric is an index used to assess the clarity of an image [29], which not only evaluates the image dehazing effect, but also illustrates the ability to maintain key features of the road surface. The basic idea is to evaluate the clarity of an image by measuring the average value of the pixel gradient in the image. In calculating the average gradient, the gradient value of each pixel in the image first needs to be calculated using the Sobel operator. Then, the gradient values of all the pixels are averaged to obtain the average gradient value. The larger the average gradient value, the higher the clarity of the image, and vice versa. The formula for calculating this metric is
(19)$$AG=\frac{1}{\left(M-1)(N-1\right)}\sum_{i=1}^{M-1}\sum_{j=1}^{N-1}\sqrt{\frac{{\left(I(i+1,j)-I(i,j)\right)}^{2}+{\left(I(i,j+1)-I(i,j)\right)}^{2}}{2}}$$
where *I* denotes the image. *M* and *N* denote the height and width of the image, respectively.

The recognition rate of road object slices: the application of road inspection images aims at extracting and discovering road disease objects. Therefore, for the images processed using different dehazing methods, the recognition rate of crack slices is examined in combination with the classical support vector machine classifier to verify the effectiveness of dehazing performance and the maintenance of disease object features.

### 4.1. Details of the Training Implementation of the Proposed Methodology

We selected the OTS (outdoor training set) and the road inspection dataset from the RESIDE dataset [19] for training, where the synthesized images of the OTS and the road inspection dataset were used for pre-training, and the real haze images from the road inspection dataset were used for fine-tuning. All images were randomly cropped into 256 × 256 sized patches, normalized from −1 to 1. In pre-training, the dehazing backbone model was trained 100 times, and the initial learning rate was set to 10^−4^ with a decay rate of 0.75. In fine-tuning, the initial learning rate was set to 10^−4^ with a decay rate of 0.5. To train the OPE module, the downsampled haze-free image and the original haze-free image were used as the training sample pairs, the learning rate was set to 10^−4^, and the batch size was set to five. Finally, the training of the proposed dehazing network was accomplished by cascading the DCSC module and OPE module.

### 4.2. Performance Analysis of Ablation Experiments with Different Modules of the Proposed Methodology

The proposed dehazing method of the DCSC-OPE-Net contains the DCSC coarse dehazing module and the OPE module. To evaluate the contribution of different modules to the proposed DCSC-OPE-Net, this section utilizes road inspection images to carry out ablation experiments to, respectively, compare the performances of using different modules, including using the traditional MSBDN module, using the DCSC-MSBDN module, using the MSBDN-OPE module, and using the whole DCSC-OPE-Net. The results are as follows (Table 1 and Figure 2):

According to the above table and figure, it can be found that the original hazed image has poor results in both the visual quality and quantitative indicators. The original image is obscured by haze, presenting a misty look. The MSBDN improves the image quality, but the results are dark overall. Applying DCSC loss, the quality of targets such as white-colored vehicles is thought to be improved, and the overall level of the image is clear. By using the OPE module, the resolution of the image is further increased and the details are enhanced. The final DCSC-OPE-Net obtains the best visual quality. From the perspective of the quantitative indicators, by applying the MSBDN dehazing module, decreases in the NIQE and BRISQUE values can be seen. When the DCSC module is added, the two metrics are further improved. It can be found that when the integrated DCSC-OPE-Net is used, better results can be obtained, which demonstrates that the different modules of the proposed method present complementary contributions to enhance the dehazing effect.

### 4.3. Evaluating the Performance of the Proposed Method Compared to SOTA Methods

To evaluate the performance advantages of the proposed method compared with the SOTA method, the performance evaluation is conducted on real haze images from public datasets and real haze images from road inspections. The information of the comparison methods is as follows:The color attention prior method (CAP): The CAP method is the classical haze-removal algorithm using a color attention prior. This method uses a linear model of colors to predict the scene depth so as to recover the haze-free image.The DCP: the DCP method is the classical dark-channel method, which serves as a representative physical modeling method for evaluating the performance of the proposed method.The Structure Representation Network (SRN) [40]: this method is a dehazing network model built in recent years, which utilizes uncertain feedback learning, thereby improving haze occlusion images with dense and non-uniform particle distributions to generate effective dehazing results.The Feature Fusion Attention Network (FFA-Net): the FFA-Net is a typical end-to-end dehazing network for the direct acquisition of haze-free images.The Principled Synthetic-to-Real Dehazing Network (PSD): the PSD network utilizes synthetic data to train real data fine-tuning to obtain effective dehazing results.

Based on these comparison methods, the network training is completed according to the same strategy as the proposed DCSC-OPE-Net, and the obtained experimental results can be found in Figure 3, Figure 4, Figure 5, Figure 6, Figure 7 and Figure 8:

According to the experimental results, it can be found that the raw images are poorly visualized, and it is difficult to clearly distinguish the objects contained in the images. Due to the patch-based assumption of the DCP, the overall brightness of its dehazing results is low. The CAP method also obtains ordinary results because the simple assumption is not proper for complex outdoor environments. Comparing the visual quality between the CAP and DCP, the CAP method obtains poor results when the color is unobvious, i.e., low contrast (see the upper one in Figure 7a). In addition, it can be found that the SRN method for dense clouds has a significant distortion in the dehazing results, especially for the sky region (see Figure 7d). Due to the processing flow of the DCP assumption, the dehazing results of the PSD method have noticeable block effects, presenting the loss of details in the image (see Figure 7e). Remarkably, the visual effect of the proposed method is better than the competitors. It is noted that the haze-free image obtained from the proposed method has a clear outline and distinctive colors. In this way, regarding different evaluation metrics, the proposed method obtains the best NIQE metrics for both the public dataset and the road inspection dataset, and its BRISQUE metrics are slightly lower than that of the PSD method.

### 4.4. Evaluating the Performance of the Proposed Method for Feature Enhancement of Road Objects

The proposed method has the ability to perceive near-view objects and enhance their edge features. To evaluate this advantage, typical road disease objects, such as road depressions, cracks, etc., cutting from haze-free images generated using different dehazing methods are selected for the comparison experiments. The extracted object slices are shown in the following figure.

As can be seen in the figure, the object of the original image is contaminated with haze and is difficult to recognize. The visualization of the diseased objects is improved by using different dehazing methods. The results of the DCP present low luminance (see Figure 9c), leading to an unobvious disease object. The results of the SRN present the distortion of colors, which may affect the detection of disease objects (see Figure 9e). The visual quality of the PSD is better than the other comparison methods, while it still lacks the rich details of disease objects (see Figure 9f). For the visual quality, it is noted that the proposed DCSC-OPE-Net dehazing method obtains the best results. Our DCSC-OPE-Net not only ensures the effective removal of haze but can also effectively recover details. Then, the following results (see Table 2 and Figure 9) are obtained from the different methods:

According to the metrics, it can be found that, after applying different dehazing methods, their metrics are improved compared to hazed images. The typical DCP method has an inherent patch-based assumption, which leads to a decrease in their object resolution and thus interferes with their evaluation metrics. The PSD method utilizes the fine-tuning of the real hazed images, which ensures better indicators for the object features. It can be found that the proposed DCSC-OPE-Net dehazing method is able to obtain the best dehazing results for the diseased objects, which will recover the clarity of the images of the road objects with a large object gradient, which indicates the effectiveness of the diseased object extraction.

To further illustrate the improvement of the proposed DCSC-OPE-Net dehazing method for disease object recognition performance, 100 samples of disease objects and non-disease objects with different dehazing methods are collected, and the typical HoG [41] feature descriptor and support vector machine [42] classifiers are utilized for the classification test of disease objects with the two-fold cross-validation strategy. The obtained classification accuracy results are displayed in Table 3.

According to the disease object classification accuracy, it can be found that the original hazed image is intricate enough to accurately achieve disease object detection and classification due to haze obscuration. After applying different dehazing methods, the results of the disease object classification are improved. The patch-based DCP method reduces their resolution and makes it difficult to accurately recognize disease objects. The CAP can obtain better results than the DCP by avoiding the confusion between a dark background and objects. Compared with the typical PSD and FFA methods, which only focus on the whole-image dehazing metrics and ignore the road objects, the proposed method can consider the whole-image dehazing performance and road object enhancement simultaneously. It is seen that the proposed DCSC-OPE-Net obtains the best accuracy in disease object classification. This finding indicates that the proposed DCSC-OPE-Net dehazing method outperforms typical dehazing methods in terms of dehazing evaluation metrics and has superior performance in road object recognition.

## 5. Discussion

Our experiments were performed on a computer with E5-2630 CPU, with an NVDIA RTX3090 GPU and the WIN10 operating system. For images with 256 × 256 resolutions, the average processing time per image of the DCSC-OPE-Net is 10.9194 s, which ensures efficient image dehazing.

Based on the overall experimental results, it can be found that the proposed DCSC-OPE-Net can obtain better haze removal results and road target recognition results. However, the proposed DCSC-OPE-Net can hardly remove the haze effectively in different environmental conditions, dataset biases, or scenarios, such as when the haze density is very large, when the signal-to-noise ratio is very low, and when the image is acquired at night. In these cases, it is difficult to remove the haze completely and recover the target information. The detailed examples can be found in Figure 10.

As can be seen from the above figure, for Figure 10a, the excessive haze density overwhelms the target, making it difficult for the proposed method to recover effective information. For Figure 10b, the dark image presents a low signal-to-noise ratio, and the dehazing effect is also poor in this case. For Figure 10c, the image is taken at night, and the road details are not obvious. It is also difficult to obtain good dehazed results in this case. Therefore, although the overall experiment results of the proposed DCSC-OPE-Net method are effective, they still have limitation conditions for application. The proposed DCSC-OPE-Net method cannot work well for heavy haze, low signal-to-noise ratios, and images taken at night.

## 6. Conclusions

To improve the degradation of road inspection image quality caused by haze occlusion, the DCSC-OPE-Net deep dehazing network method containing two parts, i.e., a DCSC coarse dehazing module and an OPE module, is proposed to remove haze and enhance road object features. Compared with the classical DCP-based methods that do not adapt to light-colored objects, the DCSC module improves on the rigid constraints of the DCP, ensuring feature retention for light-colored objects. To avoid the loss of detail information during dehazing, the OPE module is then constructed to perceive the near-view objects using an attenuation transmission map and enhance the object feature using edge gradient loss. The experiments verify that the proposed method can obtain better dehazing results (i.e., NIQE score is 4.5, BRISQUE score is 18.67, and the recognition rate is 83.67%) than the typical DCP and SOTA deep dehazing networks using different datasets. Additionally, the proposed DCSC-OPE-Net achieves the best road disease object recognition results among the comparison methods.

The proposed DCSC-OPE-Net can obtain effective results for general road inspection images. However, it may present poor results for heavy haze, low signal-to-noise ratios, and images taken at night. Therefore, our future work will develop the dark-channel module used for images taken at night and with low signal-to-noise ratios, and introduce image inpainting technology to deal with heavy haze to further ensure effective road inspection and intelligent operation work.

## Figures and Tables

**Figure 1 sensors-23-08932-f001:**
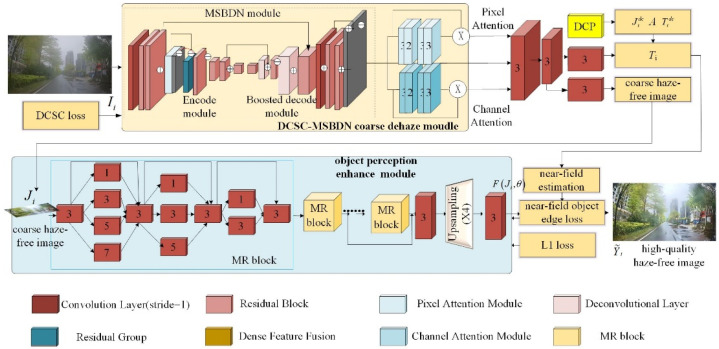
Illustration of the procedure of the proposed DCSC-OPE-Net for the haze removal of road inspection images.

**Figure 2 sensors-23-08932-f002:**
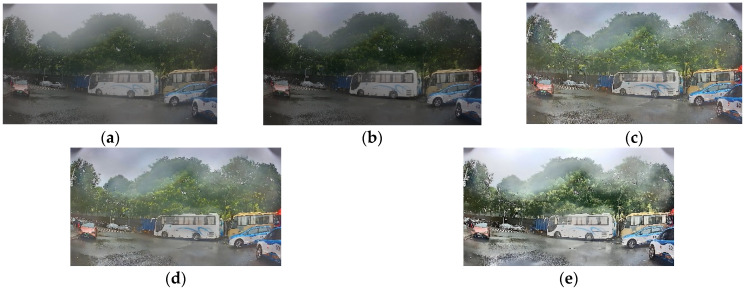
The ablation experiment results for road inspection images. (**a**) Original image. (**b**) MSBDN method. (**c**) DCSC-MSBDN method. (**d**) MSBDN-OPE method. (**e**) DCSC-OPE-Net method.

**Figure 3 sensors-23-08932-f003:**
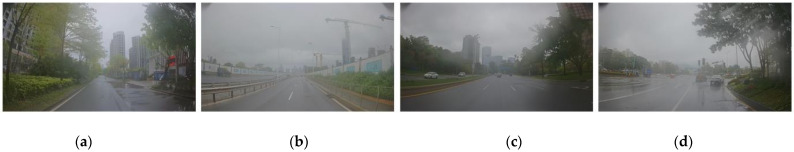
The road inspection images obscured by haze. (**a**–**d**) display haze images acquired from different roads.

**Figure 4 sensors-23-08932-f004:**
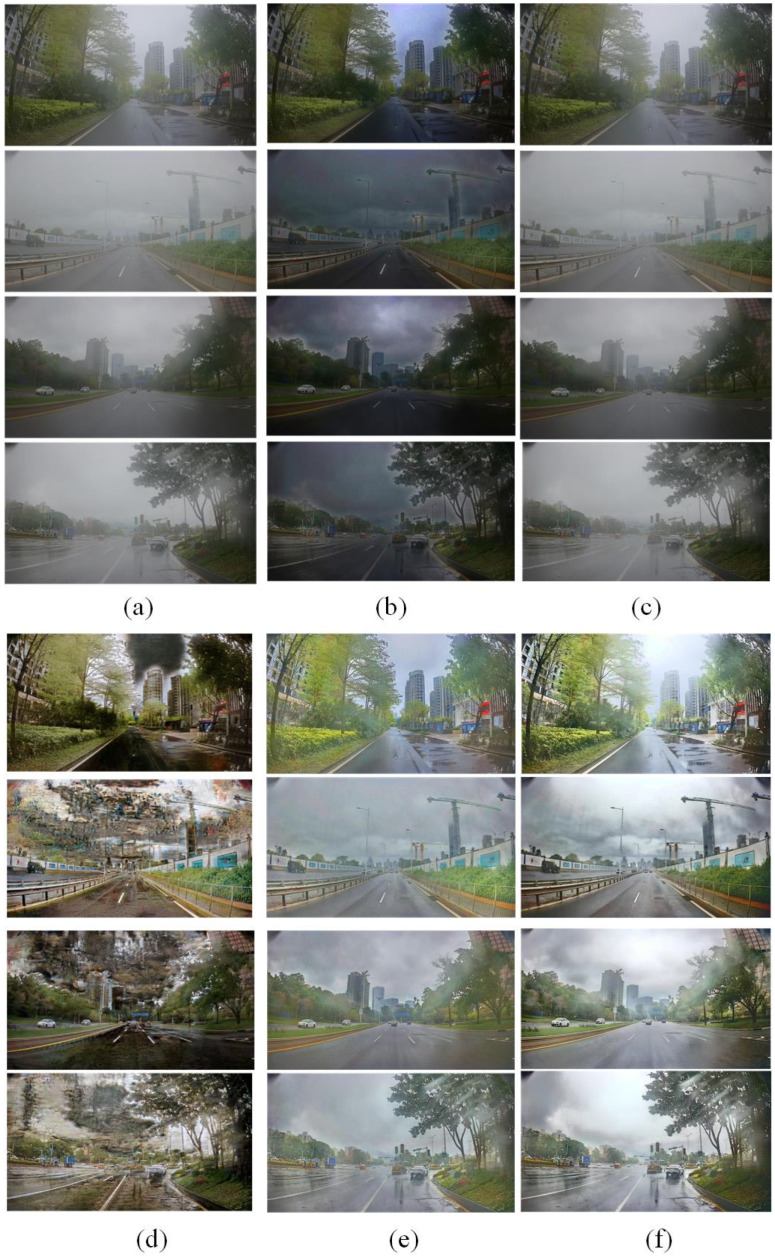
Dehazing results of road inspection images with different methods. (**a**) CAP method. (**b**) Dark-channel method. (**c**) FFA method. (**d**) SRN method. (**e**) PSD method. (**f**) The proposed method.

**Figure 5 sensors-23-08932-f005:**
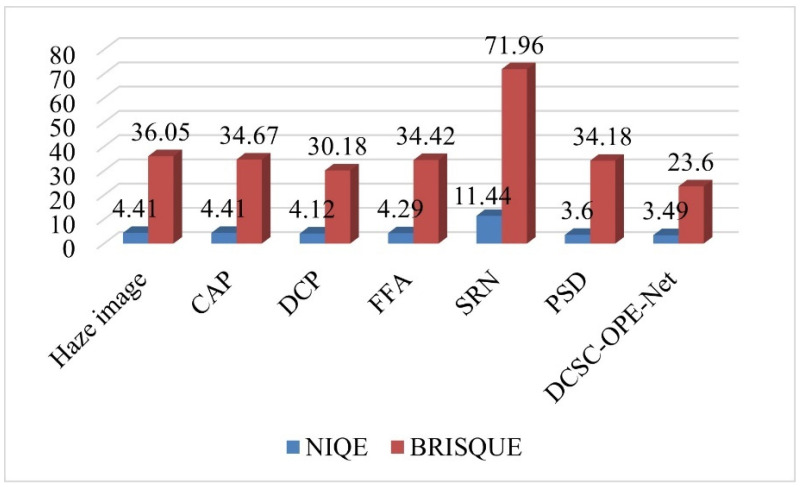
Comparison of the performance of the proposed method and the comparison methods under different metrics of the road inspection dataset.

**Figure 6 sensors-23-08932-f006:**
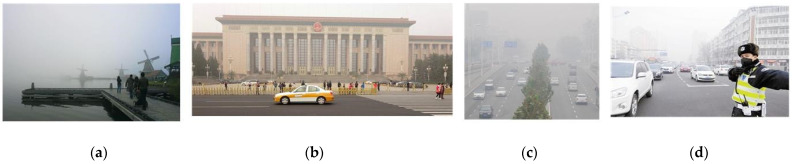
The representative images in public datasets obscured by haze. (**a**–**d**) display haze images acquired from different scenes.

**Figure 7 sensors-23-08932-f007:**
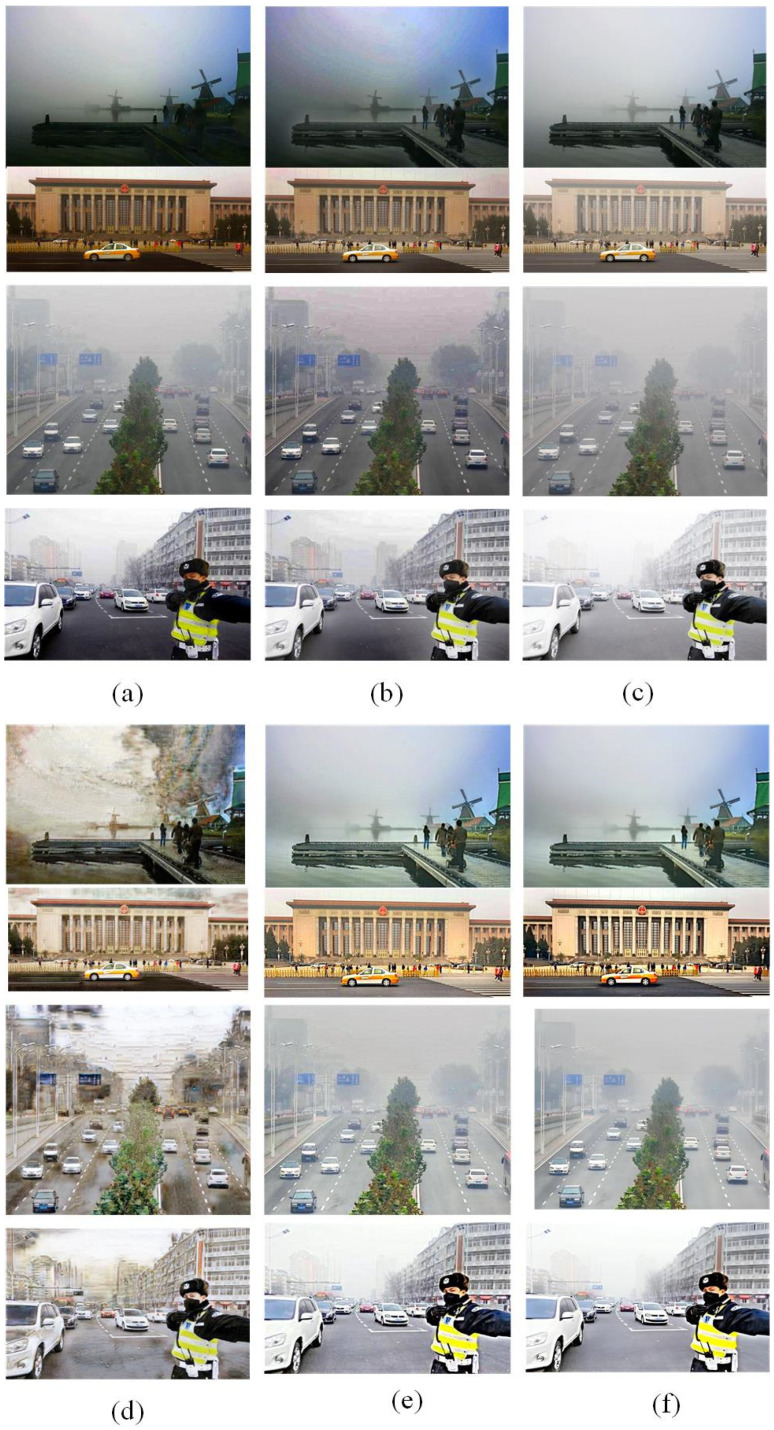
Image dehazing results of different methods for public datasets. (**a**) CAP method. (**b**) Dark-channel method. (**c**) FFA method. (**d**) SRN method. (**e**) PSD method. (**f**) The proposed DCSC-OPE-Net method.

**Figure 8 sensors-23-08932-f008:**
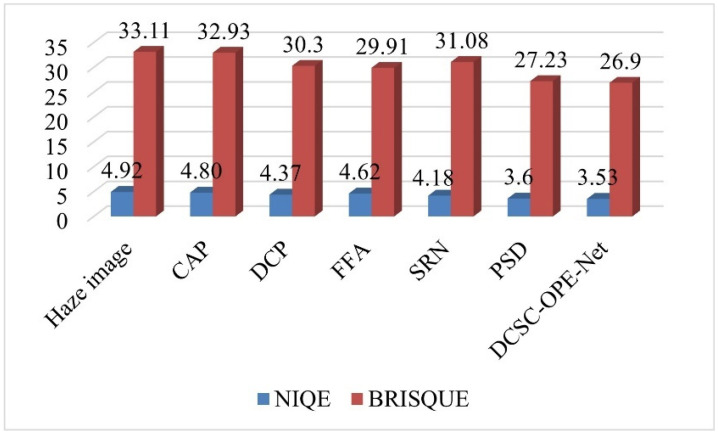
Comparison of the performance of the proposed method and the comparative methods under different evaluation metrics using the public outdoor dataset.

**Figure 9 sensors-23-08932-f009:**
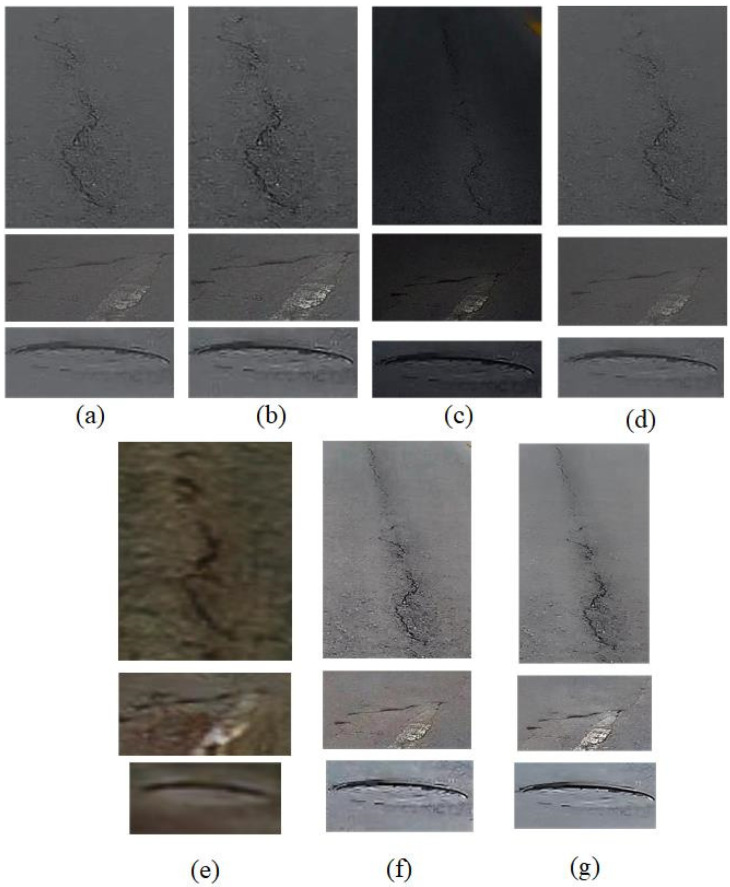
Object slices of inspection images for different dehazing methods. (**a**) Original slice. (**b**) Object slice obtained from CAP method. (**c**) Object slice obtained from DCP method. (**d**) Object slice obtained from FFA method. (**e**) Object slice obtained from SRN method. (**f**) Object slice obtained from PSD method. (**g**) Object slice obtained from the proposed DCSC-OPE-Net method.

**Figure 10 sensors-23-08932-f010:**
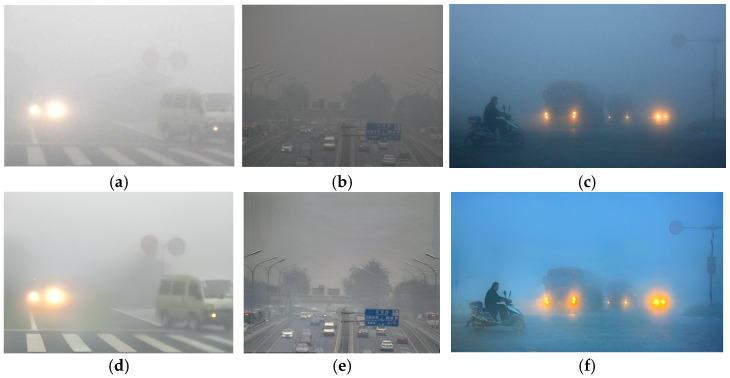
Examples of difficulties in the effective implementation of the proposed methodology. (**a**) Original image with heavy haze. (**b**) Original image with low signal-to-noise ratio. (**c**) The original image acquired at night. (**d**) Dehazed image of (**a**) using DCSC-OPE-Net. (**e**) Dehazed image of (**b**) using DCSC-OPE-Net. (**f**) Dehazed image of (**c**) using DCSC-OPE-Net.

**Table 1 sensors-23-08932-t001:** Ablation experimental results of ablation of different modules of the proposed DCSC-OPE-Net dehazing network.

	Haze Image	MSBDN	DCSC-MSBDN	MSBDN-OPE	DCSC-OPE-Net
NIQE	4.41	4.38	3.62	3.77	3.49
BRISQUE	36.05	34.19	34.2	23.4	23.1

**Table 2 sensors-23-08932-t002:** Multi-metric evaluation results of road object slices with different dehazing methods.

	Haze Image	CAP	DPC	FFA	SRN	PSD	DCSC-OPE-Net
AG	22.64	34.51	25.96	22.16	22.1	60.09	95.81
BRISQUE	8.64	8.61	8.19	8.73	19.7	6.79	4.5
NIQE	25.56	26.40	25.07	28.15	67.97	20.84	18.67

**Table 3 sensors-23-08932-t003:** Accuracy of different dehazing methods for the recognition of disease objects via SVM.

	Haze Image	CAP	DCP	FFA	SRN	PSD	DCSC-OPE-Net
recognition rate	57.0%	67.25%	55.25%	69.33%	59.17%	71.42%	83.67%

## Data Availability

Not applicable.

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
