# Peer review of "Dark-Channel Soft-Constrained and Object-Perception-Enhanced Deep Dehazing Networks Used for Road Inspection Images"

_sensors, 2023, doi:10.3390/s23218932_

Round 1
Reviewer 1 Report
Comments and Suggestions for Authors
The paper, "Dark Channel Soft Constrained and Object Perception Enhanced Deep Dehazing Networks for Road Inspection Images," addresses the issue of haze in road inspection images, which impairs the visibility of key road objects. The authors propose a novel approach, DCSC-OPE-Net, consisting of two modules: a dark channel soft-constrained dehazing module and a near-view object perception enhanced module. Unlike traditional dark channel algorithms, this approach aims to maintain the features of light-colored vehicles and lane lines effectively. It also incorporates edge enhancement and transmission map-based techniques to identify and enhance key road features autonomously. However, there are a few aspects that could be further developed.
1- While the paper mentions improved dehazing performance, it would be helpful to include visual comparisons with traditional methods to illustrate the actual quality enhancement.
2- The paper could benefit from a more detailed explanation of the proposed loss functions, as these are critical components of the network. Some readers might struggle to grasp the technicalities behind them.
3- the paper discusses the use of edge enhancement and transmission maps for object perception but lacks in-depth information on the implementation and performance impact of these techniques. Providing more insights into how these components work and their contributions to the results would enhance the paper's clarity.
4- The paper should include discussions or considerations regarding computational complexity and resource requirements for implementing DCSC-OPE-Net, especially if it's intended for real-time road inspection systems.
5- The paper highlights the superior performance of DCSC-OPE-Net in terms of dehazing and road object recognition metrics, it would be valuable to discuss the potential limitations or scenarios in which this approach might not perform optimally. Understanding the model's boundaries is essential for practical application.
6- The paper should provide more in-depth explanations of the various loss functions used, especially the dark channel soft-constrained loss. Understanding the mathematical foundations of these components is crucial for readers to grasp the methodology fully.
7- The paper highlights the superior performance of DCSC-OPE-Net in terms of dehazing and road object recognition metrics, it should also consider discussing potential limitations or scenarios in which the proposed approach might not perform optimally. Identifying these boundaries is crucial for practical use cases.
8- The paper mentions the dark channel soft-constrained loss, but the mathematical formulation and details of its implementation are not provided. It would be helpful to include equations and a step-by-step explanation of how this loss function is applied within the network.
9- The paper introduces edge enhancement and transmission maps for object perception, but the methodology for generating these maps and how they are integrated into the network remain unclear. More technical insights into these components are needed.
10- Consider including a section that discusses the potential limitations or challenges of the proposed approach, such as environmental conditions, dataset biases, or scenarios where it may not perform as effectively. Acknowledging these aspects is important for a holistic assessment of the model's applicability.
Comments on the Quality of English Language
Moderate editing of English language required
Reviewer 2 Report
Comments and Suggestions for Authors
To addresses the challenge of traditional dehazing methods in dealing with road inspection images, the paper introduces the Dark Channel Soft Constrained and Object Perception Enhanced Deep Dehazing Network (DCSC-OPE-Net). This network employs a soft constraint on the dark channel for dehazing and enhances near-view road object perception. The network also uses edge enhancement loss with a transmission map, allowing it to autonomously identify and emphasize essential road features. Through experiments on public and real-world datasets, the DCSC-OPE-Net's performance surpassed conventional dehazing networks in both dehazing quality and road object recognition metrics.
The paper is well-organized, and the experimental design is sound. The conclusions and results are clearly presented. I recommend accepting the paper for publication with some revisions.
Comments:
1) The authors' categorization of dehazing methods into "classical physical-based" and "deep learning based" is overly simplistic and can be misleading. First, dehazing techniques can be primarily divided into single-image-based and multi-image-based methods. It appears that the authors' focus is primarily on the former. This point should be clarified in the paper. Additionally, within traditional dehazing techniques, there are methods based on physical models or purely based on empirical relationships, as well as their combination. It would be beneficial for the authors to address and incorporate these distinctions in their work for a more comprehensive understanding.
2) The "Related Work" section appears to be overly concise. There have been numerous improvements based on the Dark Channel Prior (DCP), and it would be appropriate for the authors to provide a brief overview of these advancements. Additionally, there are many deep learning-based methods available. Given that the authors introduce a new network architecture in their work, it is crucial to discuss the network designs of existing methods. It would be beneficial for the authors to delve deeper into these designs and critically analyze their limitations. This would provide a more comprehensive context for the significance and novelty of the authors' work.
3) Regarding the evaluation metric, most of the studies I have reviewed typically adopt a set of metrics, retaining widely recognized measures such as SSIM, PSNR, etc. Solely using the Mean Gradient (MG) metric may provide a somewhat one-sided assessment. Furthermore, when introducing the principle and advantages of the MG metric, it is crucial for the authors to provide relevant citations.
4) In the "ablation experiments" section, it would be beneficial to include visual analyses of some typical cases. By doing so, the effectiveness of the improvements proposed by the authors can be more vividly and intuitively demonstrated. The subsequent comparison with results from other models would then showcase the superiority of the entire approach. Combining both a detailed visual analysis in the ablation section and comprehensive comparative results will make the evaluation more holistic and convincing.
5) Regarding the choice of comparison cases, I'm uncertain if there are currently any benchmark datasets available for this task. If such datasets exist, I strongly recommend the authors to utilize them. If not, it would be beneficial for the authors to provide a thorough rationale for the selection of their examples, emphasizing their representativeness and, more importantly, their relevance to the challenges put forth by the authors in the paper.
6) In the conclusion section, it would be valuable for the authors to summarize some general principles or guidelines based on their findings and methodology. Providing such overarching principles not only encapsulates the essence of their work but also offers a clear direction for future researchers in this domain.
7) If feasible, I strongly recommend the authors to openly share the code and experimental data related to this study.
Comments on the Quality of English LanguageMinor editing of English language required
Round 2
Reviewer 1 Report
Comments and Suggestions for Authors
Author's addresses all the comment and paper can be accepted in its current form.
Comments on the Quality of English LanguageMinor editing of English language required